# Succession Analysis of Gut Microbiota Structure of Participants from Long-Lived Families in Hechi, Guangxi, China

**DOI:** 10.3390/microorganisms9122524

**Published:** 2021-12-07

**Authors:** Minhong Ren, He Li, Zhen Fu, Quanyang Li

**Affiliations:** College of Light Industry and Food Engineering, Guangxi University, Nanning 530004, China; msminhong@126.com (M.R.); 1816401001@st.gxu.edu.cn (H.L.); fuzhen13@gxu.edu.cn (Z.F.)

**Keywords:** gut microbiota, longevity, centenarians, aging

## Abstract

The gut microbiota structure has been proposed to be involved in longevity. In this study, trajectories of age-related changes in gut microbiota were analyzed by comparing the gut microbiota composition from long-lived families. A specific bacterial community pattern and signature taxa of long-lived people were found in long-lived families, such as the enrichment of *Enterobacteriaceae* in all age groups and the higher abundances of *Christensenellaceae*, *Verrucomicrobiaceae*, *Porphyromonadaceae*, *Rikenellaceae*, *Mogibacteriaceae*, and *Odoribacteraceae* in long-lived elderly and the positive correlation between them. The cumulative abundance of the core microbiota was approximately stable along with age, but the genera and species in the core microbiota were rearranged with age, especially in *Ruminococcaceae* and *Lachnospiraceae*. Compared with the control group, the proportions of *Lachnospiraceae*, *Roseburia*, and *Blautia* were significantly higher in participants from the long-lived village, but their abundances gradually decreased along with age. Based on functional predictions, the proportions of pathways related to short-chain fatty acid metabolism, amino acid metabolism, and lipoic acid metabolism were significantly higher in the long-lived elderly compared with the offspring group. The trajectory of gut microbiota composition along with age in participants from long-lived families might reveal potential health-promoting metabolic characteristics, which could play an important role in healthy aging.

## 1. Introduction

Longevity has been described as the result of a complex combination of variables deriving from genetics, lifestyle, and environment [1,2]. In recent years, gut microbiota has been proposed as an important determinant of human health [3,4,5], and their composition was revealed to be involved in longevity [6,7,8,9]. The gut microbiota is highly dynamic and sensitive to environmental stimuli. The changes in microbiota composition in the human gut during the aging process, are influenced by regions, diet, genetic, age, gender, behavioral habits, and so on [10]. The microbiome constitutes the major interface between humans and the environment, which is influenced by biosocial stressors and behaviors, and it mediates effects on the health and aging process [11]. Gut microbiota are involved in substance metabolism, immune regulation, and neuromodulation of the host, which play an important role in host health [12]. Identification of gut microbiome signatures associated with longevity and modulation of the gut microbiota are rapidly emerging fields of study and hold promise for healthy aging and longevity [13,14].

Individuals who have reached very old age can be thought of as individuals whose gut microbiota has successfully reestablished a continuous relationship with the host. Healthy longevity is the best model [10] for studying the relationship between gut microbiota and aging. The study of the gut microbiota of long-lived individuals can help to understand how gut microbiota successfully adapt to age-related gradual changes in the environment (including lifestyle and diet) and maintain host health in extremely long-lived individuals. Currently, several research groups have used centenarians as a model to study aging and gut microbiota [6,15,16,17,18,19]. In 2015, research by our team revealed that the gut microbiota in centenarians was remodeled, for example, the abundance of *Faecalibacterium* and *Akkermansia* was reduced; the abundance of *Escherichia* and *Methanobrevibacter* was increased. However, the sample size of the work was small and it lacked a young control group [15]. Biagi et al. [6] found the presence of a core microbiota of highly occurring symbiotic bacterial groups, which remained approximately constant during aging but varied in the cumulative relative abundance of its members. Wu et al. [17] found that centenarians had a higher diversity of core microbiota species and microbial genes than young adults and the elderly. Tuikhar et al. [19] found higher biodiversity within *Ruminococcaceae* in centenarians, with respect to younger adults, irrespective of their nationality. The mystery of gut microbiota signatures of longevity is gradually being revealed. However, it is not clear whether these supposedly beneficial microbial communities are present throughout human life or if they disappear in normal aging and are regained by those who are long-lived.

This study took long-lived families as the research object and analyzed the gut microbiota structure of people of different ages in the long-lived families, in an attempt to provide some support for the analysis of this remodeling process. Hechi region is located in the northwest longevity belt of Guangxi (China) with Bama (the world’s fifth-highest region of longevity) as the core. It is famous for its high centenarian prevalence, and most of the local long-lived elderly are healthy. Previous research of our team found that the local long-lived elderly with the specific dietary structure of this area was correlated with the composition of gut microbiota. The long-lived elderly’s descendants who had lived with them for a long time had a highly similar environment and behavior habits (such as region, diet, work, and so on) to the long-lived elderly when they were young. In this study, people from long-lived families in Hechi were selected as the research objects. In contrast, a control group was recruited from non-long-lived families in the nearby urbanized region with low centenarian prevalence. The gut microbiota compositions were compared to find the characteristics in the change of intestinal microbiota along with age and to establish a signature taxa or bacterial community pattern specific to long-lived people.

## 2. Materials and Methods

### 2.1. Participant Recruitment and Study Groups

Participants were enrolled in the Bama, Donglan, Fengshan, and Dahua counties of the Hechi region (China). The centenarian prevalence of the four counties was 35.6, 31.4, 28.5, and 15.0 centenarians/100,000 citizens in the year of sampling, respectively. A home address information list of local long-lived families (families with people over 90 years old) was obtained from the Civil Affairs Bureau. Healthy long-lived elderly and their offspring living with them were recruited in this study. A questionnaire was administered to obtain information about age, physical condition, medical history, and dietary information from the identified participants. The anthropometric data (weight, height, and body mass index) were measured during the visit. The age of all participants was validated. In addition, participants who had medical care or received antibiotic treatment within the last six months before the fecal sample collection were excluded. In total, 31 long-lived families and 69 healthy members were enrolled in this study. Participants were divided into two groups, the LCN group (38 long-lived participants, all living at home; 90–118 years) and the LEA group (31 offspring living together with the long-lived; 38–80 years), and then four age groups, centenarians, nonagenarian, the elderly, and younger adults (Table 1 and Appendix A). Moreover, we collected 29 participants’ fecal samples from the non-long-lived families in the nearby urban region with a low centenarian ratio (Xixiangtang district of Nanning city; 0.26/100,000 citizens), as a control group (UEA group) for comparison (Table 1 and Appendix A).

This study was approved by the Ethics Committee of Guangxi University (Approval No.: GXU-M-2019003), and all participants in the study signed written informed consent.

### 2.2. Sample Collection

Fecal samples were collected in sterile stool containers with an icebox by the participants themselves or their family members and immediately stored at −20 °C. The samples were then transported to the laboratory by a team member and stored at −80 °C until analysis.

### 2.3. Microbial DNA Extraction and 16S rRNA Gene-Based Illumina MiSeq Sequencing

The microbial community DNA was extracted using MagPure Stool DNA KF kit B (Magen, Guangzhou, China) following the manufacturer’s instructions. DNA was quantified with a Qubit Fluorometer by using Qubit^®^ dsDNA BR Assay kit (Invitrogen, Carlsbad, CA, USA), and the quality was checked by running an aliquot on 1% agarose gel. Variable regions V3–V4 of bacterial 16S rRNA gene was amplified with degenerate PCR primers, 341F (5′-ACTCCTACGGGAGGCAGCAG-3′) and 806R (5′-GGACTACHVGGGTWTCTAAT-3′). Both forward and reverse primers were tagged with Illumina adapter, pad, and linker sequences. PCR enrichment was performed in a 50 μL reaction containing a 30 ng template, fusion PCR primer, and PCR master mix. PCR cycling conditions were as follows: 94 °C for 3 min, 30 cycles of 94 °C for 30 s, 56 °C for 45 s, 72 °C for 45 s, and a final extension for 10 min at 72 °C. The PCR products were purified with AmpureXP beads and eluted in the Elution buffer. Libraries were qualified by the Agilent 2100 bioanalyzer (Agilent, Santa Clara, CA, USA). The validated libraries were used for sequencing on the Illumina MiSeq platform (BGI, Shenzhen, China) following the standard pipelines of Illumina and generating 2 × 300 bp paired-end reads. Raw reads were filtered to remove adaptors and low-quality and ambiguous bases, and then paired-end reads were added to tags by the Fast Length Adjustment of Short reads program (FLASH, v1.2.11) to obtain the tags. The tags were clustered into operational taxonomic units (OTUs) with a cutoff value of 97% using UPARSE software (v7.0.1090), and chimera sequences were compared with the Gold database using UCHIME (v4.2.40) to detect. Then, OTU representative sequences were taxonomically classified using the Ribosomal Database Project (RDP) Classifier v.2.2 with a minimum confidence threshold of 0.6 and trained on the Greengenes database v201305 by QIIME v1.8.0. The USEARCH_global was used to compare all Tags back to OTU to obtain the OTU abundance statistics table of each sample. Alpha and beta diversity were estimated by MOTHUR (v1.31.2) and QIIME (v1.8.0) at the OTU level, respectively. KEGG functions were predicted using the PICRUSt2 software.

### 2.4. Statistical Analysis

Statistical analyses were performed with SPSS V22.0 statistical software for Windows (SPSS Inc., Chicago, IL, USA) to determine the statistical differences among groups using Mann-Whitney U-test (two groups) and Kruskal-Wallis test (more than two groups). A Spearman’s rank correlation was used to evaluate trends between the relative abundance of gut microbiota at family and genus level and predicted metabolic pathways. Correlations between age and the relative abundance of gut microbiota at family and genus level were determined by Pearson’s analysis with adjustment for gender and BMI. The Venn plots in OTUs were plotted with the R package “VennDiagram”version 3.1.1. Principal component analysis (PCA) in OTUs was plotted with the R package “ade4”. Other graphs were drawn by GraphPad Prism 8 (GraphPad Software, La Jolla, CA, USA).

## 3. Results

### 3.1. Fecal Microbial Community α and β Diversity in Different Age Groups from the Long-Lived Families

The gut microbiota composition of fecal samples was analyzed by performing multiplex sequencing covering the V3-V4 regions of 16S rRNA. After removing unqualified sequences, an average of 81,441 tags per sample (±7440 SD) was obtained. The microbial community richness and diversity were evaluated by the number of observed OTUs, Chao1 index, Shannon diversity index, and Simpson diversity index. Focusing on the differences between the long-lived elderly and younger adults, significantly higher α diversity was found in the long-lived elderly group (LCN group) with respect to the offspring group (LEA group) (Figure 1a). The number of observed OTUs and Chao1 index in the LCN group was significantly higher than the LEA group, although there was no significant difference in the Shannon diversity index and Simpson diversity index between the two groups. Among the four age groups, the number of observed OTUs and Chao1 index in the LN group was significantly higher than those in the LC, LE, and LA groups. There was no significant difference between the LE and LA group (Figure 1b).

The Venn diagram shows the common and specific OTUs between different groups (Figure 1c,d). There were 1115 common OTUs in LCN and LEA; 458 OTUs of LCN were absent in the LEA group. Of these, 215, 131, and 55 OTUs were classified as Firmicutes, *Bacteroidetes*, and *Proteobacteria*, respectively. It should be noted that there were some special genera in the OTUs of the LCN group, e.g., *Ruminococcaceae* (88 specific OTUs), S24-7 (32), *Erysipelotrichaceae* (17), *Christensenellaceae* (14), *Bacteroidaceae* (13), *Prevotellaceae* (13), and *Porphyromonadaceae* (11). However, there were only 110 specific OTUs in the LEA group, including 9 annotated to *Bacteroidaceae*, 8 annotated to *Prevotellaceae*, and 8 annotated to *Veillonellaceae*. According to the principal component analysis (PCA) based on the OTUs abundance profiles, the gut microbiota structure of participants in the LCN group was separated from that of participants in the LEA group (Figure 1e).

### 3.2. Comparison of Gut Microbiota between the Long-Lived Elderly Group and Offspring Group in Long-Lived Families

The gut microbiota compositional profiles for the long-lived elderly group and offspring group are shown in Figure 2. At the phylum level, the data of the top ten microorganism populations were exhibited. The gut microbiota were dominated by *Firmicutes*, *Proteobacteria*, and *Bacteroidetes*, which respectively accounted for 93.25 and 94.88% of the total sequencing number in the LCN and LEA groups. At the family and genus level, the bacteria with average relative abundance above 1% were displayed. Both groups were dominated by just four families: *Ruminococcaceae*, *Lachnospiraceae*, *Enterobacteriaceae*, and *Bacteroidaceae*, and their cumulative relative abundance was 76.01% in the LCN group and 76.54% in the LEA group. Differentiation analysis between the LCN and LEA group was conducted by the Mann-Whitney U-test with Benjamini-Hochberg correction. As shown in Table 2, *Christensenellaceae*, *Lachnospiraceae*, *Ruminococcaceae*, *Mogibacteriaceae*, *Dehalobacteriaceae* in the phylum *Firmicutes*, *Porphyromonadaceae* in the phylum *Bacteroidetes*, and *Methanobacteriaceae* in the phylum *Euryarchaeota* had significant differences between the two groups. *Christensenellaceae*, *Ruminococcaceae_Oscillospira*, and *Mogibacteriaceae* showed a significantly higher presence in long-lived elderly (*p* < 0.0001, BH corrected), whereas *Lachnospiraceae* had a lower abundance (*p* < 0.0001, BH corrected). The relative abundance of *Christensenella*, *Porphyromonas*, *Parabacteroides*, and *Anaerotruncus* was significantly higher in the LCN group than the LEA group (*p* < 0.001, BH corrected), whereas the relative abundance of *Roseburia*, *Blautia,* and *Butyricicoccus* were significantly lower in the LCN group than the LEA group (*p* < 0.001, BH corrected).

### 3.3. Function Prediction of Gut Microbiota in Long-Lived Families

The function of gut microbiota was predicted by the PICRUSt2 software using KEEG pathway categories. The relative abundances of 165 metabolic pathways at level 3 categories were obtained and compared among groups in long-lived families. There were 108 metabolic pathways with the mean abundance >0.1%, 27 of which showed significant differences between the long-lived elderly group and the offspring group (Appendix A). Among them, six metabolic pathways were involved in the carbohydrate metabolism; four pathways were involved in the metabolism of cofactors and vitamins; four pathways were involved in lipid metabolism, and four pathways were involved in amino acid metabolism. The results showed that the long-lived elderly group displayed a significant enrichment of short-chain fatty acid metabolism (propanoate metabolism and butanoate metabolism), amino acid metabolism (valine, leucine, and isoleucine degradation, glycine, serine, and threonine metabolism, lysine degradation, and tryptophan metabolism), and cofactors and vitamins metabolism (lipoic acid metabolism, ubiquinone, and terpenoid-quinone biosynthesis) gene pathways compared with the offspring group in the gut microbiota. However, the four lipid metabolism pathways (primary bile acid biosynthesis, secondary bile acid biosynthesis, sphingolipid metabolism, and glycerolipid metabolism) abundance showed enrichment in the offspring group compared with that in the long-lived elderly group. The proportion of the lipoic acid metabolism gene pathway in the long-lived elderly group gut microbiota was increased by 50% over that in the offspring group.

Spearman correlation analysis was used to reveal the relationships between the 27 gene pathways and gut microbiota at family levels. The results are exhibited by a correlation heatmap (Figure 3). The results revealed that the abundance of *Lachnospiracea* was significantly positively correlated to 14 gene pathways, including the 4 lipid metabolism pathways, 2 antibiotic synthesis pathways (biosynthesis of vancomycin group antibiotics and streptomycin biosynthesis), and 3 carbohydrate metabolism pathways (starch and sucrose metabolism, pentose phosphate pathway, and galactose metabolism). *Enterobacteriaceae* was significantly positively correlated with 11 gene pathways, including the 4 amino acid metabolism pathways (tryptophan metabolism, lysine degradation, valine, leucine, and isoleucine degradation, and glycine, serine, and threonine metabolism) and 2 short-chain fatty acid metabolisms (propanoate metabolism and butanoate metabolism). Citrate cycle, lipoic acid metabolism, valine, leucine, and isoleucine degradation gene pathways were all significantly positively correlated to *Porphyromonadaceae*, *Bacteroidaceae*, *Christensenellaceae*, *Verrucomicrobiaceae,* and Odoribacteraceae. Glycosaminoglycan degradation was significantly positively correlated with *Porphyromonadaceae*, *Bacteroidaceae*, *Odoribacteraceae*, and *Porphyromonadaceae* (r = 0.436–0.693, *p* < 0.001). Base excision repair and glycine, serine, and threonine metabolism were correlated with Christensenellaceae (r = 0.504–0.551, *p* < 0.001).

### 3.4. The Change in Fecal Bacteria Associated with Age in Long-Lived Families

The relationships between fecal gut microbiota and the host age of participants in long-lived families were analyzed by the Pearson correlation analysis. At the family level, *Lachnospiraceae* and *Prevotellaceae* were significantly negatively correlated with age, whereas *Rikenellaceae*, *Porphyromonadaceae*, *Mogibacteriaceae*, and *Odoribacteraceae* were significantly positively correlated with age (Figure 4). At the genus level, the relative abundance of *Parabacteroides* was significantly increased along with age, but *Roseburia*, *Blautia*, and *Prevotella* were significantly decreased along with age (Figure 4). In addition, the abundance of the two famous probiotics (*Lactobacillus* and *Bifidobacterium*) and two potentially beneficial bacteria (*Christensenella* and *Akkermansia*) did not significantly correlate with age (Appendix A). Interestingly, we found that *Christensenellaceae*, *Verrucomicrobiaceae*, *Rikenellaceae*, *Porphyromonadaceae*, *Mogibacteriaceae,* and *Odoribacteraceae* were significantly correlated with each other in long-lived families (Appendix A).

### 3.5. Comparison of Gut Microbiota between Long-Lived Elderly’s Offspring Group and Matched Control Group

The participants of the matched control group were recruited from the nearby urbanized region with low centenarian prevalence. Frequently detected families and genera (average proportions greater than 1.0% in all participants of any group) were displayed (Figure 5). The LEA group was dominated by four families: *Ruminococcaceae*, *Lachnospiraceae*, *Prevotellaceae*, and *Bacteroidaceae*, and their cumulative relative abundance was 73.52%. The core microbiota composition of the LEA group was different from the UEA group. Fifteen families and genera were significantly different between the two groups (Figure 6). The relative abundance of *Lachnospiraceae*, *Clostridiaceae*, *Enterobacteriaceae*, *Lactobacillaceae*, *Roseburia*, *Blautia*, *Dorea*, *Ruminococcus*, *Lactobacillus*, *Escherichia*, and *Klebsiella* were significantly higher in the LEA group than the UEA group (*p* < 0.05), whereas *Bacteroidaceae*, *Veillonellaceae*, *Megamonas*, and *Phascolarctobacterium* were significantly lower in the LEA group (*p* < 0.05). The abundance of *Porphyromonadaceae*, Rikenellaceae, *Mogibacteriaceae*, *Odoribacteraceae*, *Christensenellaceae,* and *Verrucomicrobiaceae* had no significant difference between the LEA group and UEA group (*p* > 0.05).

## 4. Discussion

The human gut microbiome is a highly diverse ecosystem, and its composition and function can be adapted to the changing conditions of the host’s life to meet the changing needs of the host. Understanding how the gut microbiota adapts to age-related changes in the host is a difficult task, and the most useful approach is a longitudinal study that monitors the trajectory of changes in long-lived people over long periods of time (decades). Such an analysis is not yet possible, because attention to the gut microbiota has only grown rapidly in recent years. Therefore, a cross-sectional study of the structure of gut microbiota in populations with different age groups is still the main method for exploring adaptive patterns of intestinal flora that vary with the age of the host. Hechi in Guangxi is a famous longevity area in China: the local long-lived elderly favor a high-fiber diet with corn and rice porridge as a staple food, complemented with a large amount of dark vegetables, such as pumpkin and sweet potato leaves [15,20]. The high prevalence of centenarians, consistent lifestyle, and low immigration rates make the region an ideal geographic area for the study of longevity. This study added a contribution to the field of aging research, by analyzing the microbiome of a particular population from long-lived families.

Gut microbiota diversity is one of the most important indicators to evaluate the intestinal microecological balance and is closely related to the health status of the host. It has been reported that reduced biodiversity of the gut microbiota was associated with disease status, including obesity [21] and autoimmune disease [22]. Our results showed that the gut microbiota of the long-lived elderly was significantly different from those of their offspring in long-lived families. The microbial community richness (OTUs numbers and Chao1 index) in the long-lived elderly was significantly higher than that of their offspring (*p* < 0.001), which was in line with previous studies [6,16,19] that compared centenarians with young adults. However, the Shannon and Simpson indices had no significant difference among centenarian, nonagenarian, elderly, and younger adult groups. OTU numbers and the Chao1 index were significantly higher in nonagenarians than centenarians (*p* < 0.05). The results might be explained by the better health status of the nonagenarians. We found nonagenarians had healthier physical conditions than centenarians during visits; most of the nonagenarians can do housework, even simple farm work.

A higher abundance of *Christensenellaceae* and *Akkermansia* in centenarians than young adults as a common distinctive gut microbiota structure has been demonstrated in both this study and previous studies from Sichuan, China, Emilia Romagna, Italy, and Manipur, India [6,16,19]. In this study, some features were noted to be unique to defined populations from the long-lived families in specific geographic locations. The relative abundances of *Porphyromonadaceae*, *Rikenellaceae*, *Mogibacteriaceae*, and *Odoribacteraceae* were significantly positively correlated with age in participants from long-lived families, which were significantly higher in long-lived people (over 90 years) than their offspring (38–80 years). *Christensenellaceae* and *Akkermansia* did not have a linear relationship with age (Appendix A), but they were highly enriched in long-lived people, more so than in their descendants. Hence, we speculate that the abundance of the two bacteria signatures of longevity might increase significantly when the elderly reach a certain age. A higher abundance of *Christensenellaceae*, *Porphyromonadaceae*, and *Rikenellaceae* have recently been found to be associated with reduced visceral adipose tissue and healthier metabolic profiles in the Italian elderly (65–79 years) [23]. The relationship between BMI index and gut microbiota was analyzed. The results showed that BMI was significantly negatively correlated with *Rikenellaceae*, *Porphyromonadaceae*, *Verrucomicrobiaceae*, and *Odoribacteraceae* (Appendix A). *Porphyromonadaceae*, *Rikenellaceae*, *Mogibacteriaceae*, *Odoribacteraceae*, *Christensenellaceae*, and *Verrucomicrobiaceae* were remarkably enriched in the long-lived elderly group, and they were significantly positively correlated with each other (Appendix A). Moreover, these specific bacteria were correlated with the significant enrichment of lipoic acid metabolism, citrate cycle (TCA cycle), valine, leucine, and isoleucine degradation, glycine, serine, and threonine metabolism, base excision repair, glycosaminoglycan degradation, and carbon fixation pathways in prokaryotes gene pathways in gut microbiota (Figure 3). Lipoic acid is a vitamin with antioxidant properties, and it is an essential cofactor of many mitochondrial enzymes involved in the pyruvic acid, alpha ketone, and glutaric acid oxidative decarboxylation of branched-chain amino acids and the cracking of glycine [24]. Lipoic acid plays an important role in intracellular glucose metabolism, which may be useful in the treatment of diabetes [25]. Therefore, we speculate that the age-related increase in *Rikenellaceae*, *Porphyromonadaceae*, *Mogibacteriaceae*, *Odoribacteraceae*, *Verrucomicrobiaceae*, and *Christensenellaceae* might contribute to lower BMI and healthier metabolic profile, which are important for healthy aging.

However, *Lachnospiraceae* and its genera *Roseburia*, *Blautia* were significantly lower in long-lived people (over 90 years) than their younger offspring (38–80 years), and they were significantly negatively correlated with age in participants from long-lived families. Meanwhile, our results displayed that the relative abundance of *Lachnospiraceae* was significantly positively correlated with carbohydrate metabolism (starch and sucrose metabolism, pentose phosphate pathway, and galactose metabolism). *Lachnospiraceae* contains many SCFAs production genera, which degrade a large array of polysaccharides, such as glucans, mannans, resistant starches, and simpler saccharides [26]. By comparing the gut microbiota of long-lived people’s offspring group (17 male, 14 female, 38–80 years, mean age 57.8 years) to that of a control group (14 male, 15 female, 28–82 years, mean age 57.9 years) from a nearby urbanized region, the relative abundance of *Lachnospiraceae* and its members *Roseburia*, *Blautia*, and *Dorea* was significantly higher in the long-lived people’s offspring group. A dietary intervention trial on a group of individuals with type 2 diabetes mellitus has demonstrated that *Lachnospiracea* increased after 12 weeks of intervention with a high-fiber diet composed of whole grains, traditional Chinese medicinal foods, and prebiotics [27]. The higher abundance of *Lachnospiraceae* in participants in long-lived villages might be attributed to a high intake of dietary fiber. Although members in the family *Lachnospiraceae* have repeatedly shown their ability to produce beneficial metabolites (SCFAs) for the host [28], the impact of human gut *Lachnospiraceae* on the host physiology is controversial [29]. Previous literature indicated the different taxa and the high abundance of *Lachnospiraceae* were associated with some diseases [30]. Our recent research found that the relative abundance of *Lachnospiracea* was significantly higher in healthy elderly than unhealthy elderly (61–80 years); *Lachnospiracea* experienced a mild decrease in healthy elderly and a significant increase in unhealthy elderly after high fiber intake with a centenarian sourced probiotics dietary intervention (unpublished). Meanwhile, the relative abundance of *Lachnospiraceae* was strongly positively correlated with antibiotic biosynthesis (streptomycin and vancomycin group antibiotic biosynthesis). The relative abundance of *Lachnospiraceae* gradually reduced along with age in healthy participants from long-lived families might be important for maintaining the balance of gut microbial community, which plays an important role in healthy aging.

Biagi et al. [6] highlighted the presence of a core microbiota of highly occurring, symbiotic bacterial groups, which remained approximately constant during aging but varied in the cumulative relative abundance of its members, and the fecal microbiota of Italians was dominated by just three families: *Bacteroidaceae*, *Lachnospiraceae*, and *Ruminococcaceae*, but their cumulative relative abundance decreased along with aging (77.8% in younger adults; 71.1% in the elderly; 58.7 ± 11.8% in centenarians; 57.7 ± 15.0% in semi-supercentenarians). The fecal microbiota in participants from long-lived families in Hechi, Guangxi was dominated by four families: *Ruminococcaceae*, *Lachnospiraceae*, *Enterobacteriaceae*, and *Bacteroidaceae*; however, their cumulative relative abundance remained stable along with aging (77.8% in younger adults; 75.2% in the elderly; 74.4% in nonagenarians; 77.4% in centenarians), in disagreement with the Italian data. Although the cumulative abundance of the core gut microbiota approximately kept stable along with age, the genera and species in the core microbiota were rearranged with age, especially in *Ruminococcaceae*. The relative abundance of *Oscillospira* (belonging to the family *Ruminococcaceae*) was significantly higher in long-lived people than their offspring, in line with the result reported by Biagi [6]; whereas, *Faecalibacterium* was significantly lower in long-lived people. The low abundance of *Faecalibacterium* in centenarians was also detected in other studies [6,7,16,17,18,31,32], suggesting that it could be part of the aging process itself, regardless of lifestyle and dietary habits. In addition, there were 88 specific OTUs annotated to Ruminococcaceae only detected in nonagenarians and centenarians, suggesting that the *Ruminococcaceae* diversity was higher in long-lived people.

Members of *Proteobacteria* generally are Gram-negative facultative anaerobes, most of which are pathogenic bacteria. *Proteobacteria* is prevalent in the air, water, and soil (77.9%, 61.3%, and 36.5% respectively), and usually have a small proportion (about 4.5%) in the human gut [33]. Studies have shown a bloom of *Enterobacteriaceae* associated with metabolic endotoxic septicemia and inflammatory bowel diseases [34,35]. Previous studies revealed that the abundance of *Proteobacteria* in the centenarians were higher than that of younger adult [18,19]. Yu et al. [36] also found that the abundance of *Proteobacteria* in participants from the longevity area (Gaotian Village, Liuyang, Hunan, China) was higher than that of the non-longevity control area (13.78% vs. 3.40%). We previously found that the proportion of *Proteobacteria* in centenarians from Bama county was as high as 21.5% [15]. In this study, we also found a high abundance of *Proteobacteria* in healthy people from long-lived families in Hechi, Guangxi (27.1% in centenarians; 23.7% in nonagenarians; 17.3% in elderly; and 17.2% in young adults), which was significantly higher than that of a control group in Nanning, Guangxi (8.7%). Interestingly, we found *Enterobacteriaceae* was significantly positively correlated with amino acid metabolism gene pathways (lysine degradation, tryptophan metabolism, valine, leucine, and isoleucine degradation, and glycine, serine, and threonine metabolism) and short-chain fatty acid metabolism (butanoate metabolism and propanoate metabolism). A previous study found SCFA production via fermentation of amino acids such as L-lysine fermentation butanoate [17]. In addition, many members of *Enterobacteriaceae* are facultative anaerobes, whereas most microbes in the gastrointestinal tract are obligate anaerobes. This unique oxygen requirement of Proteobacteria might influence the relation between the abundance of *Proteobacteria* and oxygen homeostasis or concentration in the gastrointestinal tract. The facultative anaerobes in *Proteobacteria* (e.g., *Escherichia*, *Klebsiella*, and *Enterobacter* species) make the habitat suitable for colonization by strict anaerobes, by consuming oxygen, altering the pH, lowering the redox potential, and producing carbon dioxide and nutrients [37]. Thus, the high abundance of *Proteobacteria* and *Enterobacteriaceae* might play a positive role in the health of participants in the long-lived families.

## 5. Conclusions

This study revealed some specific trajectories of age-related changes in gut microbiota by comparing the gut microbiota composition in participants from long-lived families with a wide age span (38–118 years). A specific bacterial community pattern and signature taxa specific to long-lived people were found in long-lived families, such as the enrichment of *Enterobacteriaceae* in all age groups and the higher abundances of *Christensenellaceae*, *Verrucomicrobiaceae*, *Porphyromonadaceae*, *Rikenellaceae*, *Mogibacteriaceae*, and *Odoribacteraceae* in long-lived elderly. The cumulative abundance of the core gut microbiota was stable along with age, but the genera and species among the core microbiota were profoundly rearranged with age, especially in *Ruminococcaceae* and *Lachnospiraceae*. Compared with an urban control group, the proportions of *Lachnospiracea* and its members (*Roseburia* and *Blautia*) were significantly higher in participants from the long-lived village, but the abundances of *Lachnospiraceae*, *Roseburia,* and *Blautia* gradually decreased along with age in long-lived families. Based on the functional predictions, the result revealed that the gut microbiota in the long-lived elderly had a higher capacity for short-chain fatty acid metabolism, amino acid metabolism, and lipoic acid metabolism compared with their younger offspring. The trajectory of gut microbiota composition along with age in participants from long-lived families revealed the potential health-promoting metabolic characteristics, which could play an important role in healthy aging. Future research should combine gut microbiota with multiple omics data to explore more detailed mechanisms of the relationship between gut microbiota and healthy aging.

## Figures and Tables

**Figure 1 microorganisms-09-02524-f001:**
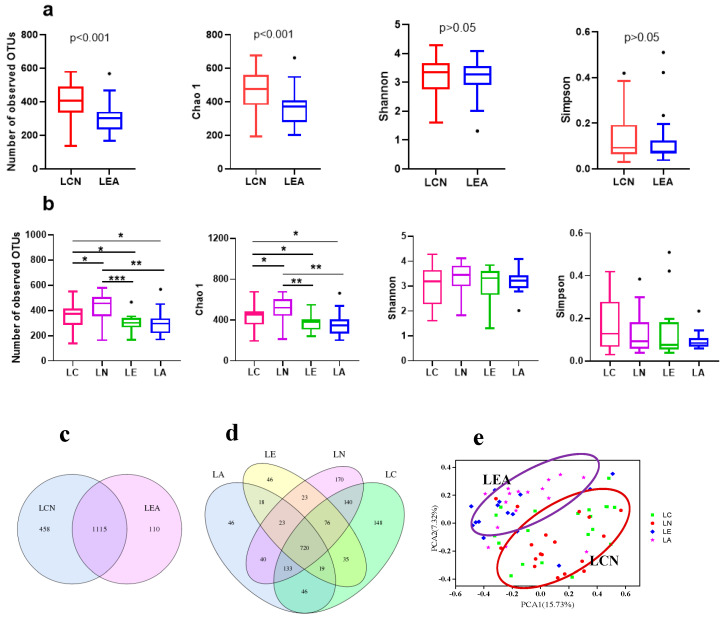
Comparison of microbial community diversity among groups in long-lived families. (**a**) Comparison of observed OTUs, Chao1, Shannon index, and Simpson index between the LCN and offspring groups in long-lived families; (**b**) Comparison of observed OTUs, Chao1, Shannon index, and Simpson index among the four age groups: centenarians, nonagenarians, elderly, and younger adults in long-lived families; (**c**) Venn diagram showing the unique and shared OTUs between the LCN and LEA groups; (**d**) Venn diagram showing the unique and shared OTUs among the four age groups; (**e**) PCA plot generated by using the abundance profile of OTUs. * *p*-value < 0.05; ** *p*-value < 0.01; *** *p*-value < 0.001, “·“ indicates outlier.

**Figure 2 microorganisms-09-02524-f002:**
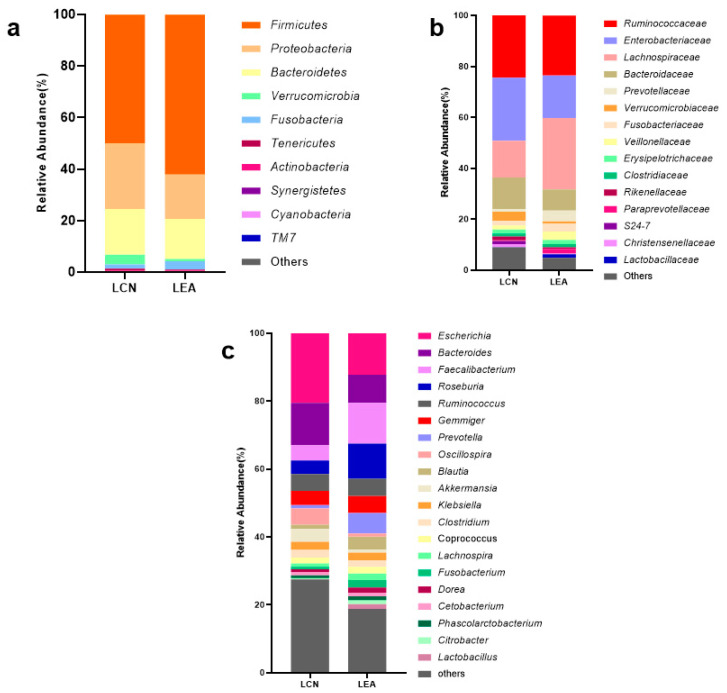
Bar chart of the relative abundance (%) gut microbiota profiles at the phylum (**a**), family (**b**), and genus (**c**) level. At the phylum level, the top ten taxa are exhibited. At the family and genus level, the taxa with average relative abundance above 1% are displayed.

**Figure 3 microorganisms-09-02524-f003:**
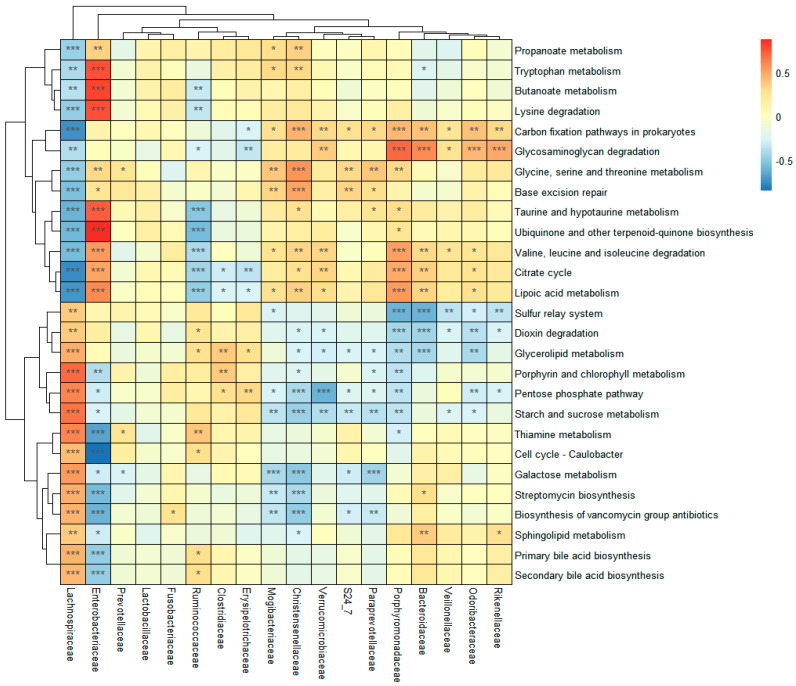
Heat map of the Spearman rank correlations between the relative abundance of predicted metabolic pathways and gut microbiota at the family level. Red indicates a positive correlation and blue indicates a negative correlation; * *p*-value < 0.05; ** *p*-value < 0.01; *** *p*-value < 0.001.

**Figure 4 microorganisms-09-02524-f004:**
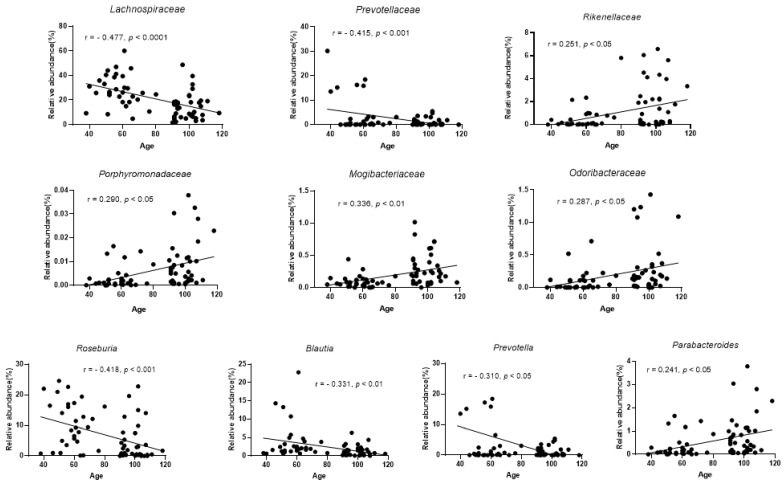
The families and genera significantly correlated with age. Rank tests with Pearson’s correlation coefficient were used to assess the correlations between gut microbiota and age with 69 participants in long-lived families (adjusted with gender and BMI).

**Figure 5 microorganisms-09-02524-f005:**
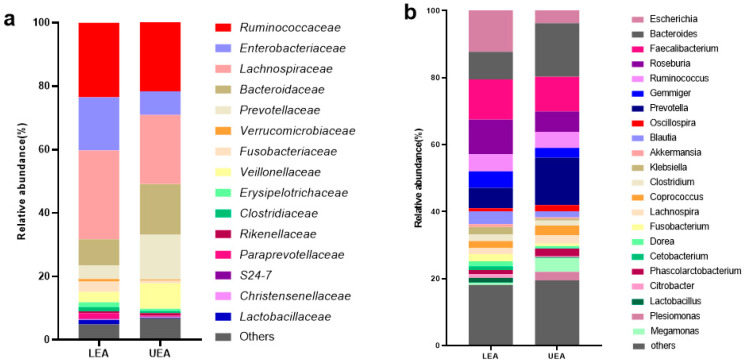
Bar chart of the relative abundance (%) gut microbiota profiles at family (**a**) and genus (**b**) level. The taxa with average relative abundance above 1% were displayed.

**Figure 6 microorganisms-09-02524-f006:**
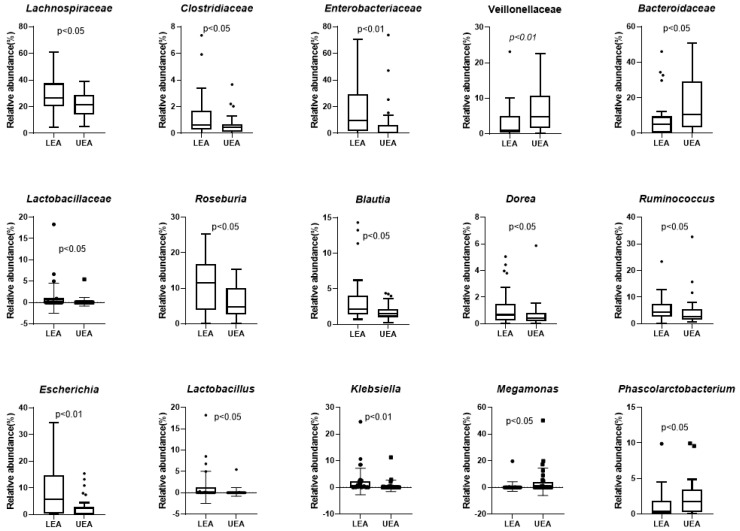
The families and genera with significant differences between the LEA and UEA groups. “·” indicates outlier. “▪” indicates outlier.

**Table 1 microorganisms-09-02524-t001:** Demographic and anthropometric characteristics in research and control groups.

Parameters	Research Groups				Control Group
	LCN Group		LEA Group		UEA Group
	Centenarian (LC) (*n* = 20)	Nonagenarian (LN) (*n* = 18)	Elderly (LE) (*n* = 15)	Adult (LA) (*n* = 16)	Elderly and Adult (*n* = 29)
Age	104 ± 4 (100–118)	93 ± 2 (90–98)	66 ± 6 (60–80)	50 ± 6 (38–58)	58 ± 15 (28–82)
Female/Male	18/2	15/3	6/9	8/8	15/14
Height (cm)	141.5 ± 6.2	145.4 ± 8.4	156.3 ± 10.3	157.9 ± 6.4	160.4 ± 7.4
Weight (kg)	39.1 ± 4.9	39.7 ± 6.4	53.5 ± 14.0	58.1 ± 10.3	61.8 ± 7.6
Body mass index (BMI, kg/m^2^)	19.4 ± 2.2	18.8 ± 2.4	21.1 ± 3.8	23.2 ± 3.2	22.7 ± 2.9

All values are presented as mean ± SD.

**Table 2 microorganisms-09-02524-t002:** Gut microbiota significantly differed between the long-lived elderly group and the offspring group.

Taxa	Mean Relative Abundance (%)		Log2 Fold Change ^1^	*p*-Value (BH Corrected)
	LCN	LEA		
*Firmicutes*				
*Christensenellaceae*	1.21	0.34	1.85	0.0001
*Christensenellaceae_Christensenella*	0.02	0.00096	4.36	0.002
*Lachnospiraceae*	14.66	28.46	−0.96	0.0001
*Lachnospiraceae_Roseburia*	4.04	10.42	−1.37	0.0032
*Lachnospiraceae_Blautia*	1.28	3.88	−1.59	0.004
*Lachnospiraceae_Epulopiscium*	0.065	0.00043	7.25	0.011
*Ruminococcaceae_Oscillospira*	5.02	0.97	2.37	0.0002
*Ruminococcaceae_Anaerotruncus*	0.034	0.01	1.74	0.008
*Ruminococcaceae_Butyricicoccus*	0.19	0.55	−1.53	0.002
*Ruminococcaceae_Faecalibacterium*	4.64	11.98	−1.37	0.051
*Clostridiaceae_Sarcina*	0.038	0.00054	6.12	0.012
*Mogibacteriaceae*	0.29	0.089	1.68	0.0005
*Dehalobacteriaceae*	0.018	0.0041	2.12	0.009
*Dehalobacteriaceae_Dehalobacterium*	0.018	0.004	2.16	0.011
*Proteobacteria*				
*Desulfovibrionaceae*	0.44	0.23	0.92	0.017
*Oxalobacteraceae*	0.012	0.0039	1.59	0.009
*Oxalobacteraceae_Oxalobacter*	0.12	0.0038	1.64	0.011
*Peptococcaceae*	0.019	0.0089	1.09	0.038
*Bacteroidetes*				
*Odoribacteraceae*	0.29	0.097	1.57	0.010
*Odoribacteraceae_Odoribacter*	0.16	0.032	2.27	0.022
*Porphyromonadaceae*	0.90	0.33	1.43	0.009
*Porphyromonadaceae_Porphyromonas*	0.87	0.0016	5.75	0.003
*Porphyromonadaceae_Parabacteroides*	0.80	0.32	1.34	0.004
*Rikenellaceae*	1.57	0.57	1.46	0.037
*Verrucomicrobia*	3.7	0.85	2.12	0.050
*Euryarchaeota*	0.012	0.00067	4.16	0.014
*Methanobacteriaceae*	0.012	0.00076	4.00	0.009
*Methanobrevibacter_Methanobrevibacter*	0.012	0.00081	3.92	0.011
*Lentisphaerae*	0.0058	0.003	0.95	0.049
*Synergistetes*	0.16	0.038	2.07	0.048

^1^ Fold change in the relative abundance of taxa in the LCN group in comparison to the LEA group.

## Data Availability

The data in this study are available on request from the corresponding author.

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
