# Peer review of "Succession Analysis of Gut Microbiota Structure of Participants from Long-Lived Families in Hechi, Guangxi, China"

_microorganisms, 2021, doi:10.3390/microorganisms9122524_

Round 1

Reviewer 1 Report

Minhong Ren et al report age-related changes in gut microbiota by comparing the gut microbiota from longevity families. Although the study and its finds are of interest, the work lack details that are required for microbiome studies. The manuscript lacks clarity in writing and can be improved.  

Major concerns:

Lack of detailed statistical analysis reporting. Missing confounder analysis especially diet-related which authors have collected but not utilized. Also, gender can have an influence on the microbiome in the elderly. 

"The data in this study are available on request from the corresponding 
author" At least the raw data with age information needs to be publically accessible without having to contact the corresponding author. 

Change subjects to participants
Line 20-21 what is the meaning of highly detected?
Line 31 GM full-form?
Line 43: Best model? Reference for this?
Line 90: delete big.

Sorry for the comment on the English writing but in many instances, the statements are grammatically incorrect and sometimes not clear what the authors mean to say.

Some other comments

Line 144: normalized sequence depth? What is the depth? please give details.
Figure 1e is this tested with PERMANOVA ? The authors state in the legend that PCA biplot generated by using the abundance profile of OTUs?
This figure does not look like a biplot. Did they use relative abundance or count abundance. Was it euclidean or another distance?

The methods section needs a much more detailed description.

Did the authors check for zero inflation in data? Why use Pearson instead of Spearman?
References are not as per journal requirements.

The discussion and conclusions need to be put in perspective with the limitations of the study which is currently lacking.

Author Response

All comments of the reviewer were responded below in blue. Almost all suggestions were accepted and included in the revised manuscript (highlighted in red).

Reviewers' comments:

Reviewer 1:

Comments to the Author

Minhong Ren et al report age-related changes in gut microbiota by comparing the gut microbiota from longevity families. Although the study and its finds are of interest, the work lack details that are required for microbiome studies. The manuscript lacks clarity in writing and can be improved.

Response: Thank you very much for your attention on the manuscript. We appreciate your positive comments and sincerely accept your comments on this manuscript. We have polished the manuscript by English Editing serve and the modifications are highlighted in red in the revised manuscript.

Lack of detailed statistical analysis reporting. Missing confounder analysis especially diet-related which authors have collected but not utilized. Also, gender can have an influence on the microbiome in the elderly.

Response: Thank you very much for your useful comments. We would supplement the detailed statistical analysis data, but the data is too large to include in the manuscript as supplement table. According to your comments, we have adjusted for confounding factors for gender and BMI. (Please seen in section 2.4 and Figure 4 in the revised manuscript)

"The data in this study are available on request from the corresponding

author" At least the raw data with age information needs to be publically accessible without having to contact the corresponding author.

Response: Thank you very much for your useful comments. We have added the raw data with age information. (Please seen in Tabel S1 in the revised manuscript).

Change subjects to participants

Response: Thank you very much for your useful comments. We have changed “subjects” into “participants”. (Please seen lines 2, 18, 22, 78-79, 86, 88, 90, 94, 103, 253, 256, 270, 272, 304, 326, 354, 367, 379, 409-410, 430, 442, 447, 454, 471, 478 in the revised manuscript)

Line 20-21 what is the meaning of highly detected?

Response: Thank you very much for your useful comments. We have corrected the description. (Please seen lines 20-21 in the revised manuscript)

Line 31 GM full-form?

Response: Thank you very much for your useful comments. The “GM” has been changed into “gut microbiota”. (Please seen line 31 in the revised manuscript)

Line 43: Best model? Reference for this?

Response: Thank you very much for your useful comments. The reference [10] has been added. (Please seen line 43 in the revised manuscript)

Line 90: delete big.

Response: Thank you very much for your useful comments. The “big” has been deleted. (Please seen line 91 in the revised manuscript).

Sorry for the comment on the English writing but in many instances, the statements are grammatically incorrect and sometimes not clear what the authors mean to say.

Response: Thank you very much for your useful comments. We have polished the manuscript by English Editing serve and the modifications are highlighted in red in the revised manuscript.

Some other comments

Line 144: normalized sequence depth? What is the depth? please give details. Figure 1e is this tested with PERMANOVA ? The authors state in the legend that PCA biplot generated by using the abundance profile of OTUs?

This figure does not look like a biplot. Did they use relative abundance or count abundance. Was it euclidean or another distance?

Response: Thank you very much for your useful comments. The “normalized sequence depth” has been deleted and the detailed sequence information has been added. (Please seen lines 148-150);

The “PCA biplot” has been changed into “PCA plot” and the PCA plot is not involved in PERMANOVA test and distance; PCA plot generated by using the count abundance of OTUs. (Please seen Figure 1 in the revised manuscript)

The methods section needs a much more detailed description.

Response: Thank you very much for your useful comments. We have added some more detailed description. (Please seen the section 2.1 and 2.4 in the revised manuscript)

Did the authors check for zero inflation in data? Why use Pearson instead of Spearman?

Response: Thank you very much for your useful comments. As far as we know it's needed to check for zero inflation, when the data has a lot of zeros. In the correlation analysis, we selected high-abundance bacteria, and observed intuitively the data with very few 0 values, so there was no check for zero inflation. Pearson is mainly used for linear correlation analysis, so Pearson correlation was used to analyze the correlation between age and intestinal flora, while Spearman was used for others.

References are not as per journal requirements.

Response: Thank you very much for your useful comments. The format of reference has been corrected. (Please seen the section References in the revised manuscript)

The discussion and conclusions need to be put in perspective with the limitations of the study which is currently lacking.

Response: Thank you for your attention on the manuscript. We have modified the discussion and conclusions. (Please seen Lines 290, 293, 299, 314, 320, 326-329, 332, 353-354, 383, 396-397, 447-451 in the revised manuscript)

Reviewer 2 Report

The research is well written , specially for graphics 

Author Response

Reviewer 2

Comments and Suggestions for Authors

The research is well written , specially for graphics 

Response: Thank you very much for your positive comments.